# Innovative Antifungal and Food Preservation Potential of *Eucalyptus citriodora* Essential Oil in Combination with Modified Potato Peel Starch

**DOI:** 10.3390/foods14040602

**Published:** 2025-02-12

**Authors:** Nabila Khan, Saeeda Fatima, Muhammad Bilal Sadiq

**Affiliations:** Kauser Abdullah Malik School of Life Sciences, Forman Christian College (A Chartered University), Lahore 54600, Pakistan; 235683714@formanite.fccollege.edu.pk (N.K.); 21-10100@formanite.fccollege.edu.pk (S.F.)

**Keywords:** modified starch, *Eucalyptus citriodora*, antifungal, preservation, radial growth inhibition, coatings

## Abstract

The quest for natural preservation systems is on the rise due to health hazards associated with synthetic preservatives. The current study explores a significant research gap by investigating the antifungal potential of *Eucalyptus citriodora* essential oil (EO) and modified potato peel starch (M-PPS)-based coatings to control the fungal contamination in peanut kernels, providing a sustainable food preservation system. Potato peel starch was extracted by water steeping and modified by autoclaving. *E. citriodora* EO was extracted from leaves by steam distillation and chemically characterized using a gas chromatography mass spectrometer (GC-MS). The antifungal potential of EO was evaluated by radial growth inhibition assay (RGI). EO completely inhibited the growths of *Penicillium griseofulvum* and *Aspergillus niger* at a concentration of 3.125 µL/mL, which was marked as the minimum inhibition concentration (MIC). EO induced cellular leakage from fungal cells, and hyphae became pitted, indicating the strong antifungal potential of EO. EO (2 ×MIC)-treated rice seeds showed complete inhibition of *A. niger* after 7 days of incubation, and in the control treatments, all the rice seeds were contaminated (100% contamination index). M-PPS and EO-based coatings controlled the growth of *P. griseofulvum* in peanut kernels. After incubation for 7 days, control treatments were fully contaminated with fungal growth, whereas the M-PPS and EO-based coatings restricted the growth of fungi in peanut kernels. The M-PPS and EO-based preservation system can be used for the preservation of perishable food products.

## 1. Introduction

Potato is one of the most significant crops worldwide, due to its vast utilization in food-based products [1]. Food processing industries, including fast-food chains and restaurants, as well as the domestic side, widely utilize potato in various forms, such as French fries, chips, curries, hash browns, and baked or mashed potatoes [2]. During potato processing, peeling is an important step that results in the generation of potato peel waste (PPW). Based on peeling techniques (lye peeling, extruded peeling, steam peeling, or abrasive peeling), around 15% to 40% of the proportion of potato tubers accounts for PPW [1,3]. Due to limited utilization, peels are dumped into the environment and serve as a substrate for microorganisms, causing greenhouse gas emissions. By 2030, it is expected that approximately 8000 kilotons of PPW could be produced, resulting in 5 million tons of CO_2_-equivalent greenhouse gas emissions upon its disposal [4].

PPW is gaining popularity among the food scientists due to its good nutritional profile, such as phenolic compounds, dietary fibers, minerals (phosphorous, calcium, magnesium, and iron), and vitamins (B-complex and vitamin C) [3,4]. Additionally, researchers are attracted to the valorization of PPW because of its richness in valuable byproducts, such as biopolymers (lignocellulose and starch), production of green products (biofuels, bio-sorbents, biochar, and biofertilizers), and extraction of several biochemicals (zeaxanthins, violaxanthins, and lutein), enzymes, and organic acids [4,5].

Due to the increasing focus on environmental protection, sustainability, and consumer demand to reduce chemical usage, a call for an ecofriendly, biodegradable, and natural food preservation system has been fueled [6]. Among biopolymers, starch is extensively used in various industries. For the sustainable and economical production of starch, beyond the conventional sources (corn, potatoes, rice, wheat, and pulses), there is a gap in understanding the potential of non-conventional sources, particularly agrifood waste [7,8]. Potato peels are a biodegradable and sustainable source of starch [9]. The starch in potato peels accounts for 52% of dry weight, with minute amounts of fermentable sugars, i.e., 0.6% of dry weight [3,6]. Due to the presence of several hydroxyl groups, starch in its native form possesses restricted functionalities (weak mechanical characteristics and high water vapor permeability), making it an unsuitable candidate for the formulation of biodegradable packaging systems. Therefore, various techniques (physical, chemical, enzymatic, and genetic) are employed to modify native starch (NS) into resistant starch (RS) to achieve optimal characteristics [9]. Chemical modifications may introduce toxic residues. On the contrary, physical modifications are considered as green modification techniques and have wide acceptability among the consumers [10]. Autoclaving (hydrothermal treatment) is one of the physical modifications that governs the principle of repeated heating (usually above 100 °C under high pressure) and cooling cycles (at 4 °C) to enhance functional properties [11].

Biodegradable coatings act as a protective barrier between food commodity and environmental stressors, such as moisture, oxygen, and microbial contamination. Such packaging systems not only enhance the shelf life of food commodities but also reduce health hazards associated with the leaching of chemicals from synthetic packaging [12]. There are several bioactive agents (essential oils, plant-based extracts, and metallic nanoparticles) incorporated in biodegradable food packaging to augment its bioactivity [13]. Essential oils (EOs) are considered an active agent in food coatings and packaging due to their broad-spectrum biological activities [14]. However, the volatility of EOs is one of the major challenges that restrict its applications. Biodegradable coatings are utilized as an encapsulating system for the sustained delivery of EOs. This possesses synergistic functionalities, where the bioactive potential of EOs complement the physical barrier properties of biodegradable coatings [15]. *Eucalyptus citriodora* belongs to the Myrtaceae family of Eucalyptus. *E*. *citriodora* EO is known for its versatility as an antimicrobial, antioxidant, anti-inflammatory, and anticarcinogenic agent. *E*. *citriodora* EO has fascinated food and pharmaceutical industries due to its bioactive diversity [16].

Despite the growing interest in natural preservation systems, to the best of our knowledge, there are very limited data available on the development of modified potato peel starch and *E*. *citriodora* EO-based preservation systems. Our study addresses a significant gap in literature and aims to investigate the potential of combining *E*. *citriodora* EO and modified potato peel starch to formulate an innovative preservation system.

## 2. Materials and Methods

### 2.1. Sample Collection

Fresh potato peels were collected from local street food vendors of Lahore, and *E. citriodora* leaves were collected from the botanical garden of Forman Christian College (A Chartered University), Lahore.

### 2.2. Preparation of Modified Starch

The starch was extracted from potato peels using the water steeping method, as per previously published protocol [8]. Potato peels were washed, cut into small pieces, and blended to make a slurry. Afterwards, distilled water was added to the potato peel slurry (1:2 *w*/*v*) and thoroughly homogenized, followed by settling for 4 h. The slurry obtained was passed through the sieve (pore size 80 μm), and the residue collected on the sieve was further washed with distilled water to extract the remaining starch. The filtrate was stored for 24 h at 4 °C for starch settling. Subsequently, the supernatant was decanted, and multiple washings were performed by using distilled water to remove the remaining impurities. Afterwards, the starch was dried at 45 °C for 48 h in a drying oven. The dried starch was ground and stored in a zip lock bag for further use.

Native potato peel starch (N-PPS) was modified by heat treatment by following [17]. N-PPS was suspended in water (1:10 *w*/*v*) and pre-gelatinized at 85 °C for 30 min. The pre gelatinized starch was autoclaved at 135 °C for 30 min, followed by refrigeration for 24 h at 4 °C. The heating and cooling cycle was repeated thrice to obtain modified potato peel starch (M-PPS). Finally, it was dried in a drying oven at 50 °C and stored in a zip lock bag for further analysis.

### 2.3. Characterization of Starch

#### 2.3.1. Color Analysis

The color of the N-PPS and M-PPS was measured by using a handheld colorimeter (CR-10, Minolta, Osaka, Japan). CIE Lab system (L*, a*, b*) was used to express the results [18].

#### 2.3.2. Microstructure Analysis

The granular morphologies of N-PPS and M-PPS were observed under the light microscope by following [17]. The starch sample (20 mg) was suspended in 2 mL of distilled water. A microscopic slide was prepared by adding a drop of suspension on it and was observed under the light microscope (Meiji Techno, Saitama, Japan) at a magnification of 40×.

#### 2.3.3. FTIR (Fourier-Transform Infrared Spectroscopy)

Chemical fingerprinting of N-PPS and M-PPS was analyzed using an FTIR spectrometer (Agilent Technologies, Santa Clara, CA, USA) in the range of 500 to 4000 cm^−1^ wavenumbers [8].

#### 2.3.4. Thermo-Gravimetric Analysis

Thermo-gravimetric analysis (TGA) of N-PPS and M-PPS was performed as described by [19]. The samples were heated from 30 °C to 600 °C at the rate of 20 °C/min under the nitrogen atmosphere using the thermo-gravimetric analyzer (SDT Q600, Ta Instruments, Champaign, IL, USA). The total weight loss of the starch samples was measured in relation to the increase in temperature.

### 2.4. Extraction of E. citriodora EO

EO from *E. citriodora* leaves was extracted by steam distillation [20]. The leaves were washed and placed into the steam distillation apparatus for 8 h. EO was collected in an airtight, amber-colored glass bottle and stored at 4 °C for further use.

### 2.5. Characterization of E. citriodora EO

#### 2.5.1. Chemical Composition of Essential Oil

Chemical composition of *E. citriodora* EO was determined by the GC-MS system (GC-7890A/MS-5975C, Agilent Technologies, Santa Clara, CA, USA) equipped with an HP-5 MS capillary [21]. For ionization mode, the high-energy electron (70 eV) system was utilized. Helium gas was used as a carrier gas (1.0 mL/min), and the total run time was 30 min. The data were acquired by collecting the full-scan mass spectra within the range of 50–600 amu.

#### 2.5.2. Antifungal Activity

The antifungal activity of *E. citriodora* EO was evaluated against *Penicillium griseofulvum* (OR484895) and *Aspergillus niger* (MN786323) by radial growth inhibition (RGI) assay [22]. Potato dextrose agar (PDA) plates were prepared by adding twofold concentrations of EO (100, 50, 25, 12.5, 6.25, 3.125, 1.56, and 0.78 µL/mL), along with Tween 80 (2% of EO *v*/*v*). A fungal plug of 8 mm (5–7-days-old fungal culture) was placed in wells made with sterile cork borers in PDA plates. The PDA plate containing Tween 80 only was used as a control. All petri plates were incubated at 28 °C for 7 days. Vernier caliper was used to measure the fungal radial growth, and RGI (%) was determined by Equation (1). The minimum inhibitory concentration (MIC) was recorded as the lowest concentration of EO that inhibited the fungal growth until the 7th day of incubation.(1)$$RGI\left(\%\right)=\frac{dc-dt}{dc}\times 100$$
where dc and dt represent the diameters of control and EO treatment, respectively.

#### 2.5.3. Effect of EO on Fungal Hyphae

The effects of EO on fungal hyphae of *P. griseofulvum* and *A. niger* were determined by following the method described by [22]. Fungal spores (10^4^ spores/mL) were prepared and inoculated (100 μL) on PDA plates. The plates were incubated at 28 °C for 24 h. A small plug (8 mm) from the inoculated plate was placed into 20 mL of sterile potato dextrose broth (PDB) and incubated in a shaking incubator at 30 °C, 150 rpm, for 48 h. The fungal hyphae were harvested by centrifugation at 4000 rpm for 5 min, followed by washing twice with phosphate buffer saline (PBS, pH 7.4). Then, the fungal plugs were separately resuspended in 20 mL of PBS, along with EO at a concentration of MIC and Tween 80 (2% of EO *v*/*v*). PBS containing fungal plug and Tween 80 was used as the control. All the samples were further incubated at 30 °C, 150 rpm, for 24 h, followed by the staining of hyphae with methylene blue, and observed under a light microscope.

#### 2.5.4. Effect of EO on Leakage of Fungal Cellular Constituents

The effect of EO on the cellular leakage of *P. griseofulvum* and *A. niger* was observed by following [23], with slight modifications. Fungal spores (10^4^ spores per mL) were inoculated in PDB. The freshly harvested fungal cell pellets (48 h grown in PDB) were washed twice with 5 mL of PBS (pH 7.4) by centrifugation for 5 min at 8000 rpm. The fungal cell pellets were separately resuspended in 20 mL of PBS, along with different concentrations of EO (1/2 × MIC, MIC, 2 × MIC) and Tween 80 (2% of EO *v*/*v*). PBS containing Tween 80 was used as a control. All the samples were placed in a shaking incubator at 30 °C, 150 rpm, for 24 h. A sample of 3 mL was withdrawn at different time intervals (0, 2, 4, 20, and 24 h), and centrifugation at 10,000 rpm for 10 min was performed. A UV-visible spectrophotometer was used to measure the optical density of the resultant supernatant at 260 nm.

### 2.6. Preservation of Rice Grains by E. citriodora EO

The antifungal effect of *E. citriodora* EO was determined in rice grains against *A. niger* by following [24], with a few modifications. The grains were autoclaved and sprayed with *A. niger* spores (10^4^ spores/mL for 30 grains). A sterilized filter paper was dampened with sterile distilled water (1 mL) and placed in the petri plate. Then, the spore-inoculated rice grains (30 grains) were placed on each plate. Different concentrations of EO (1/2 × MIC, MIC and 2 × MIC) were prepared in 10% dimethyl sulfoxide and Tween 80 (2% *v*/*v* of EO). From each concentration, 2.5 mL was sprayed on the spore-inoculated rice grains. The spore-inoculated grains without EO were used as the control treatment. The plates were incubated for 7 days at 28 °C, and after the incubation period, 10 grains from each treatment were shifted to freshly prepared PDA plates, which were further incubated for 24 h at 28 °C. The contamination index (CI%) was calculated by using the number of infected grains out of the total grains per treatment. All treatments were performed in triplicate.

### 2.7. Preservation of Peanuts by Modified Potato Peel Starch and E. citriodora EO-Based Coatings

M-PPS and *E. citriodora* EO-based hydrogel coatings were prepared as described previously [25], with slight modifications. M-PPS (4 g) was dissolved in 100 mL of distilled water, followed by heating at 90 °C for 30 min. In the gelatinized starch, 30% glycerol (*w*/*w* on the basis of the dry weight of starch) was added and further mixed for 3 min. The solution was cooled to room temperature and homogenized (Talboys, Bio force, Irvine, UK) with *E. citriodora* EO, along with Tween 80 (0.25% *v*/*v* of EO), at 4000 rpm for 2 min to achieve coarse emulsion. The resultant coarse emulsion was sonicated at 50% amplitude for 10 min to achieve uniform coating (Talboys, Bio force, UK).

The preservation potential of EO and modified starch-based coatings on peanut kernels was evaluated against *P. griseofulvum* by following [26], with slight modifications. Peanut kernels (250 g) were surface-sterilized with sodium hypochlorite (2% *v*/*v*), followed by washing with autoclaved water. The sterilized kernels were exposed to fungal spores (10^4^ spore/mL) for 2 min, followed by drying. The peanuts were treated with five different coating treatments containing fixed concentrations of starch (4% *w*/*v*) and glycerol (30% *w*/*w*), followed by varied concentrations of EO (0%, 1.25%, 2.5%, and 5% *v/v*) and Tween 80 (0.25% *v*/*v* of EO). The coating without EO was used as control 1, while peanuts kept without any coatings were used as control 2. The peanuts inoculated with spores (15 kernels in each group) were dipped into their respective coatings for 2 min and allowed to dry at room temperature. Next, they were placed in petri plates and stored at 28 °C for 7 days. After the incubation period, three grains from each treatment were shifted to freshly prepared PDA plates, and a contamination index (CI%) was determined after further incubation for 24 h at 28 °C. All treatments were performed in triplicates.

### 2.8. Statistical Analysis

All experiments were performed in triplicates of independent experiments (except starch color analysis, performed in 5 replicates), and significant differences (*p* < 0.05) among mean observations were highlighted by using one-way analysis of variance and Tukey’s HSD tests. The instrumental analyses, such as FTIR, TGA, and GC-MS, were performed once.

## 3. Results

### 3.1. Starch Extraction

PPS was extracted using the water steeping method, and a yield of 5.1% (wet basis) was obtained, which was higher than the previous report in [19], which reported approximately 4% starch in potato peels.

### 3.2. Characterization of Starch

#### 3.2.1. Color Analysis

Based upon the CIE and the L*, a*, b* system, color attributes of N-PPS and M-PPS were observed (Appendix A). The L* value (lightness) of M-PPS (50.18) was significantly more decreased than the N-PPS (79.26) when subjected to autoclaving.

#### 3.2.2. Microstructure Analysis

The alteration in microstructures of N-PPS and M-PPS was observed by using a light microscope. The starch granules in N-PPS were in their intact regular form, with smooth surfaces. However, in M-PPS, the starch granules lost their structural integrity, with amorphous rough surfaces being more prominent (Appendix A).

#### 3.2.3. FTIR (Fourier-Transform Infrared Spectroscopy)

FTIR spectra were observed to evaluate the changes in the chemical fingerprinting of N-PPS and M-PPS. The peak associated with the O-H bond at 3253 cm^−1^ in N-PPS was shifted to 3264 cm^−1^ in M-PPS. This major shift for the O-H group of N-PPS was attributed to autoclaving, resulting in structural alteration. The C-H bond was allocated to the peak at 2925 cm^−1^ in N-PPS, which shifted to 2920 in M-PPS [27]. Another major shift was observed for the bond C=O and C-C-C being associated with functional groups from 1336 cm^−1^ (N-PPS) to 1352 cm^−1^(M-PPS) (Figure 1).

Peaks between 980–1040 cm^−1^ corresponded to the anhydrous glucose ring, and major peak shifts were observed within this region after the modification of starch [17]. There was a sharp decrease in the intensity of the peak at 998 cm^−1^ after the modification, which was attributed to a change in the structural configuration of starch. It was previously reported that changes in the configuration of glycosidic linkage were responsible for variations in the IR region in the range of 920–1000 cm^−1^ [28].

#### 3.2.4. TGA (Thermo-Gravimetric Analysis)

TGA analysis was carried out to determine the thermal stability of N-PPS and M-PP. The thermal degradation was displayed in two successive regions (Figure 2). Around 280 °C (region I), weight reduction was observed due to moisture loss. An abrupt weight loss was seen from 280 °C to 340 °C (region II) because of the thermal decomposition of starch. This might be associated with starch pyrolysis and the breakdown of inter- and intramolecular bonds. The starch was completely degraded around 586 °C, leaving behind residual mass (carbonaceous amorphous residue) [19].

### 3.3. Composition of E. citriodora EO

GC-MS analysis revealed the presence of 11 chemical compounds in *E. citriodora* EO (Table 1 and Appendix A). The major chemical compounds present in *E. citriodora* EO were citronellal (58.214%) citronellol (13.805%), 2,6-octadiene, 2,6-dimethyl (9.790%), caryophyllene (8.271%), and eucalyptol (4.610% of total), respectively.

### 3.4. Antifungal Activity

The antifungal activity of *E. citriodora* EO was studied by measuring the RGI (%) against *P. griseofulvum* and *A. niger* (Figure 3, Appendix A). The fungal growths of *P. griseofulvum* and *A. niger* remained completely inhibited at the concentration of 3.125 µL/mL, which was marked as the MIC.

### 3.5. Effect of EO on Fungal Hyphae and Cellular Leakage

The effect of EO on the fungal hyphae was observed in terms of breakage and lesions in hyphae after treatment with the MIC. In the control treatment, hyphae were intact; however, after EO treatment, hyphae were pitted, and lesions were observed under the microscope, which indicated the compromised integrity of hyphae (Figure 4).

The effect of EO was observed upon the leakage of cellular constituent from fungal mycelia. There was an increase in the absorbance, with increases in the incubation time and concentration of EO (Figure 5).

### 3.6. Preservation of Rice Grains by E. citriodora EO

EO-treated rice grains inhibited the growth of *A. niger* in comparison to the control treatment. After the 3rd day of incubation, the control treatment (rice grains without EO) showed fungal growth in a few rice grains, and on the 7th day, all the rice grains in the control treatment were contaminated in comparison to EO-treated rice grains, in which no visible contamination was found (Appendix A). After the 7th day, rice grains from each treatment were transferred to PDA plates, and after 24 h of incubation, the contamination index (CI) in the control treatment was 100%; in the highest tested EO concentration, no contamination was observed (0% CI). The ½ MIC and MIC treatments showed CIs of 30% and 15%, respectively (Figure 6).

### 3.7. Preservation of Peanuts by Modified Starch and E. citriodora EO-Based Coatings

M-PPS and EO-based coatings controlled the growth of *P. griseofulvum* in peanut kernels. After incubation for 7 days, control treatments were fully contaminated with fungal growth, whereas the M-PPS and EO-based coatings restricted the growth of fungi in peanut kernels (Appendix A). However, only a few peanut kernels were contaminated in the coating treatment with the lowest EO concentration (1.25%). After the 7th day, peanuts from each treatment were transferred to PDA plates, and after 24 h of incubation, the CI in control treatments was 100%, and the coating of M-PPS with the highest test EO concentration (5%) showed no evidence of fungal growth (CI = 0%). However, the coatings of M-PPS with 1.25% and 2.5% showed CIs of 88.9% and 22.22%, respectively (Figure 7).

## 4. Discussion

The starch modification alters the physicochemical characteristics and their techno-functional properties. During starch modification, at a high temperature (135 °C), non-enzymatic browning such as caramelization is initiated [29]. Furthermore, the color change acts as a quick indicator for structural modifications and physicochemical transformation. The autoclaving process performed under high temperature and pressure results in the breakdown of amylose–amylopectin chains. This causes the generation of short chains, particularly reducing sugars. Under repeated cooling–heating cycles, these compounds impart a characteristic yellow to brown color, which results in the lowering of L* values [29]. Therefore, in the current investigation, the color attribute (L* value) of M-PPS was lower than that of N-PPS. Furthermore, the structural integrity of M-PPS was compromised during thermal modification, which was evident by rough, amorphous surfaces in M-PSS. At a higher temperature, the disruption of the interaction between starch chains causes the structural deformation of granules [17].

The FTIR spectra indicated a change in structural configuration after modification. The functional groups were assigned to characteristic peaks by following the previously published literature [19]. There was no significant difference in the chemical functionalization observed between native and modified starch. When starch was primarily subjected to chemical modifications (acetylation, acid hydrolysis, oxidation, or etherification), significant differences were observed by the loss of previous bands or the addition of new bands [30,31]. In this study, there was no new functional group introduced, and the consistent peaks suggested that the fundamental chemical structures of native and modified starch largely remained unchanged during autoclaving. Autoclaving is a physical modification method in which starch is treated at a high temperature and pressure. Therefore, this process particularly induces physical changes, such as granular deformation, color changes, or gelatinization. This remained evident, particularly through changes in peak intensity. The intensities of several peaks were decreased in M-PPS, which indicated an alteration in structural arrangement after modification.

With respect to thermal stability, M-PPS exhibited slightly better thermal properties than N-PPS. During thermal modification, amylose molecules are released from starch chains. The released amylose molecules reform strong amylose–amylose interactions because of its linear structure, which might be associated with better thermal stability in M-PPS [32].

The bioactive potential of EOs is majorly influenced by its chemical composition. Monoterpenoids are the major constituents of *E. citriodora* EO [33], and 22 chemical compounds were identified in *E. citriodora* EO, with the most dominant being citronellal (48.7% of total), followed by citronellol (20.2% of the total). In the current study, citronellal was the predominant component (58.2%), which was higher than in the previously published report. The difference in chemical composition might be relevant to the maturity stage of leaves, EO extraction methods, geographical distribution, and environmental and genetic conditions [9,14]. *E. citriodora* EO revealed exceptional antifungal activity at lower concentrations. Souza et al. [34] also reported 100% mycelial growth inhibition of *P. digitatum* by direct contact and volatile action of *E. citriodora* EO at a concentration of about 2 µL/mL. *E. citriodora* EO was reported to completely inhibit the growth of seven phytopathogenic fungi at a concentration of 4 µL/mL, and the antifungal potential was correlated with the presence of terpenes in the EO [33]. Terpenes, due to their polar nature, can penetrate fungal membranes and induce disruption by increasing the fungal membrane permeability and fluidity [35]. EOs are reported to hinder the ergosterol biosynthesis pathway, which affects the permeability and fluidity of fungal cell membranes. As a result, structural modifications in fungal hyphae occur, including curling, twisting, and collapsing; thus, hyphal elongation stops [36]. In this study, visual observation under the light microscope confirmed the chitin degradation, as the blue dye that stains the chitin in the fungal cell wall was faint in EO-treated hyphae [23]. Moreover, the bioactive compounds present in EOs produce an antifungal effect by interacting and damaging the cell wall and cell membrane, as well as DNA, disturbing cell permeability and resulting in cellular content leakage [37].

Primarily, post-harvest losses are associated with fungal contamination. The antifungal effect of *E. citriodora* EO on rice grains was evaluated in a dose-dependent manner. At low concentrations, the bioactive components of EO exhibited sub-lethal effects on the fungi, and due to limited nutritive conditions, the fungal growth might not be supported in rice grains. Upon shifting to PDA, the nutrient-rich environment was provided to the fungi. The rice grains treated with the highest EO concentration did not even display fungal growth on PDA, indicating a stronger antifungal impact in a dose-dependent manner. However, the growth was potentiated in rice grains treated with ½ MIC and MIC, suggesting partial damage to the fungal spores, which might be recovered upon the provision of favorable conditions [38]. In the control group, there was no protective barrier; therefore, the fungal growth was triggered in either a limited nutritive environment or under nutrient-rich conditions.

The high volatility rate of EO is a major challenge for the sustained release of bioactive compounds. Biodegradable food coatings can be utilized as a sustainable delivery system. In the current research, M-PPS and *E. citriodora* EO-based coatings were applied on the peanut kernels. The M-PPS based coatings encapsulated *E. citriodora* EO, which might contribute to the sustained release of bioactive compounds [16]. High concentrations of EO induced significant oxidative stress and enhanced the antifungal efficacy in a dose–response relationship. Therefore, at the highest concentration, the inhibitory effect of EO was maintained for a prolonged period. On the contrary, at low concentrations, the threshold amount of bioactive compounds might not be available to suppress the fungal growth for a persistent timespan [37]. Additionally, the M-PPS-based coatings were homogenously dispersed over the peanut kernels and served as protective barrier against fungal spores by reducing the moisture permeability [39]. M-PPS and *E. citriodora* EO exerted synergistic effects in controlling the contamination index in peanut kernels.

Based on the promising findings, M-PPS and *E. citriodora* EO-based coatings could be utilized to tackle broader industrial challenges. The bioactive potential of such preservation systems would not only extend the shelf life of the fresh produce but also reduce post-harvest losses and synthetic plastic landfills. Additionally, the valorization of agrifood waste and the production of biodegradable packaging would align with the circular economy and green businesses principles.

Despite the current investigation presenting new avenues for the food preservation system, certain limitations need to be addressed to improve the industrial applicability of the developed coatings. Future studies should explore coating parameters, such as thickness, homogeneity, and stability, to obtain consistent coating quality. In this study, the coatings were not evaluated in real-world storage conditions, such as fluctuations in temperature, humidity levels, and storage time period, which might act as critical parameters in the efficacy of the food preservation system. Moreover, the coatings were aimed for direct application on food surfaces. However, future studies could explore the application of M-PPS and *E. citriodora* EO-based thin films and assess mechanical properties (tensile strength) and barrier properties (oxygen permeability and water permeability). Another key limitation was the evaluation of the encapsulation efficiency, loading capacity, and release kinetics of EO, confining understanding of sustained release characteristics. Addressing these gaps in future studies will pave the way to enhance the practicality and applicability of eco-friendly coatings for broader commercial use.

## 5. Conclusions

The EO of *E. citriodora* has a strong antifungal effect at a very low concentration, with the MIC being 3.125 µL/mL against two major foodborne fungi (*P. griseofulvum* and *A. niger*). *E. citriodora* EO, along with the modified starch-based coating, inhibited the growth of fungi in peanuts artificially inoculated with fungi. The high cost of natural preservative systems is always an obstacle in the sustainable application of food preservation systems. Potato peel starch, due to its low cost, and *E. citriodora* EO, due to its abundance, can be used to formulate a low-cost and sustainable food preservation system for the food and feed industry. 

## Figures and Tables

**Figure 1 foods-14-00602-f001:**
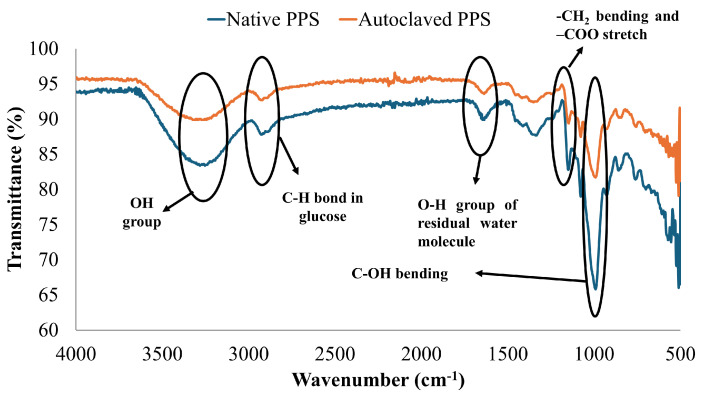
FTIR spectra of native and modified potato peel starch.

**Figure 2 foods-14-00602-f002:**
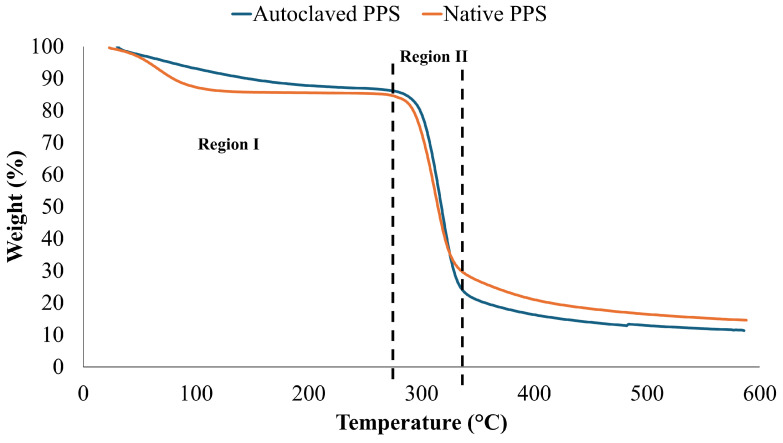
Thermo-gravimetric analysis of native and modified potato peel starch.

**Figure 3 foods-14-00602-f003:**
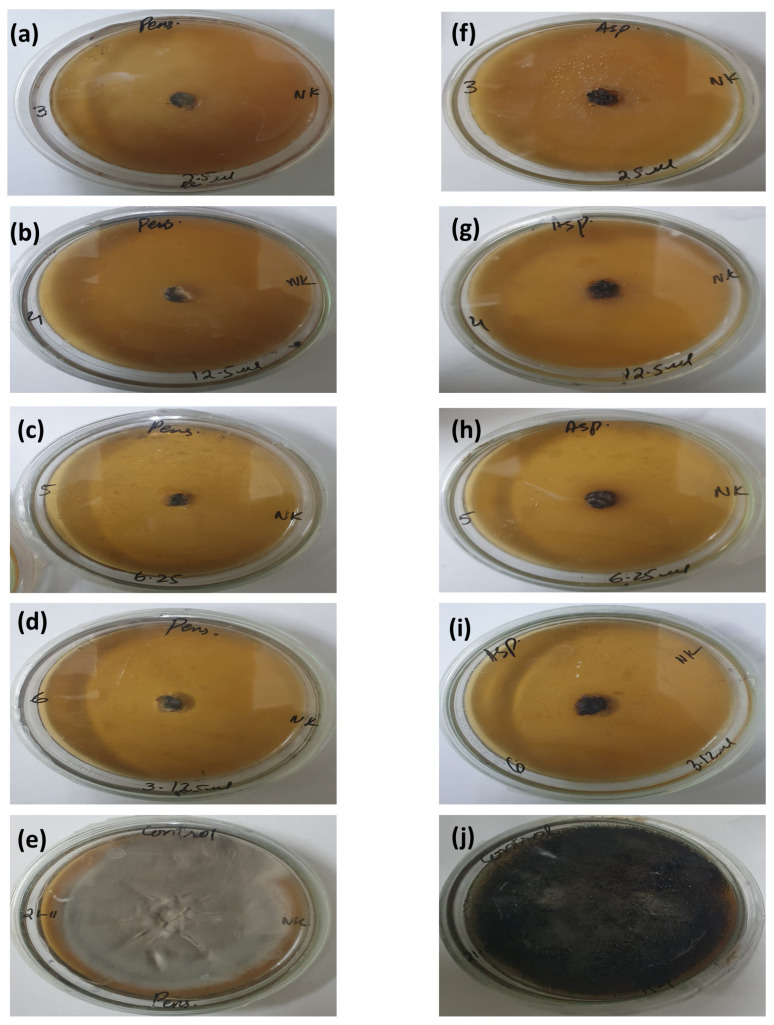
Antifungal effect of *E. citriodora* EO against *A. niger* and *P. griseofulvum;* (**a**–**e**) represent the radial growth inhibition of *P. griseofulvum* by EO, whereas (**f**–**j**) present radial growth inhibition of *A. niger*.

**Figure 4 foods-14-00602-f004:**
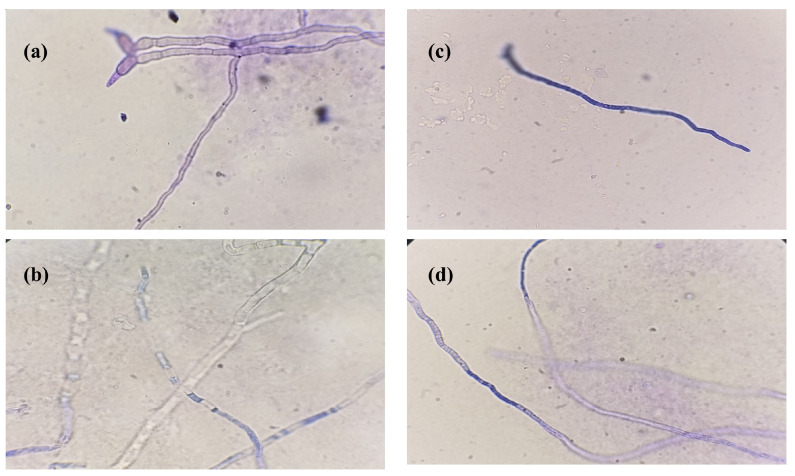
Effect of EO on fungal hyphae; (**a**,**c**) represent the hyphae of *A. niger* and *P. griseofulvum* without EO treatment, respectively, whereas (**b**,**d**) correspond to the hyphae of *A. niger* and *P. griseofulvum* after EO treatment.

**Figure 5 foods-14-00602-f005:**
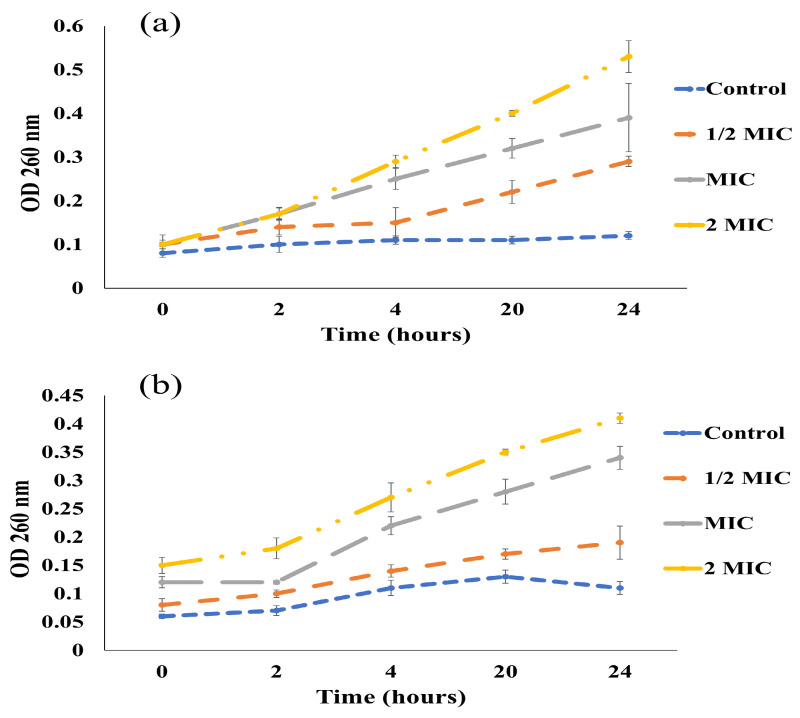
Effect of EO on the cellular leakage of fungi; (**a**) *A. niger* and (**b**) *P. griseofulvum*.

**Figure 6 foods-14-00602-f006:**
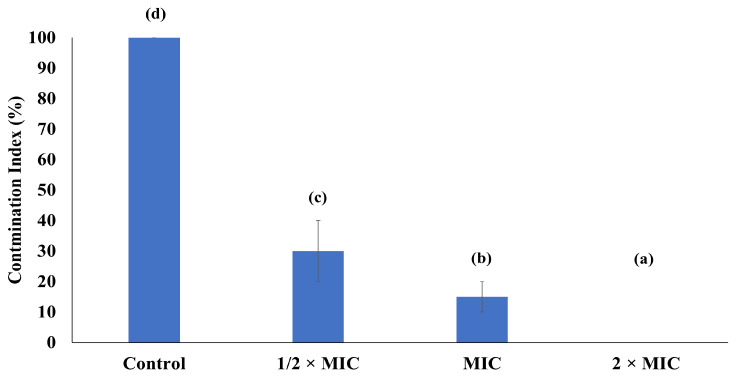
Control of fungal contamination by *E. citriodora* EO in rice grains inoculated with *A. niger.* Different superscript small letters indicate mean observations that are significantly different; 1/2 × MIC = 1.56 μL/mL, MIC = 3.125 μL/mL, and 2 × MIC = 6.25 μL/mL.

**Figure 7 foods-14-00602-f007:**
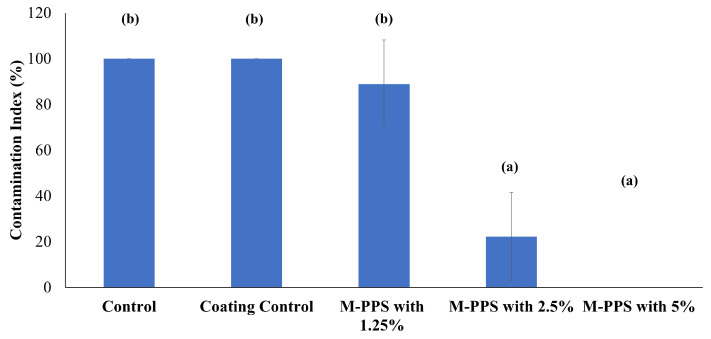
Control of fungal contamination in peanuts by modified potato peel starch and *E. citriodora* EO-based coating. Different superscript small letters indicate mean observations that are significantly different. M-PPS corresponds to modified potato peel starch containing different concentrations of EO (1.25, 2.5 and 5%).

**Table 1 foods-14-00602-t001:** Chemical composition of *E. citriodora* EO determined by GC-MS.

Peak #	Retention Time (mins)	% of Total	Compound	Molecular Formula
1	6.848	1.236	Cyclohexene, 4-methylene-1-(1-methylethyl)	C_10_H_16_
2	7.688	0.778	Bicyclo[3.1.0]hex-2-ene, 4-methyl-1-(1-methylethyl)	C_10_H_16_
3	8.733	4.610	Eucalyptol	C_10_H_18_O
4	9.199	0.714	(+)-4-Carene	C_10_H_16_
5	9.718	1.315	(+)-4-Carene	C_10_H_16_
6	11.005	58.214 *	Citronellal *	C_10_H_18_O
7	11.198	0.724	Cyclohexanol, 5-methyl-2-(1-methylethenyl)	C_10_H_18_O
8	12.062	13.805	Citronellol	C_10_H_20_O
9	13.760	9.790	2,6-Octadiene, 2,6-dimethyl	C_10_H_18_
10	14.793	8.271	Caryophyllene	C_15_H_24_
11	15.192	0.543	Humulene	C_15_H_24_

* The component present in the highest concentration (in terms of peak area).

## Data Availability

The original contributions presented in the study are included in the article/Appendix A, further inquiries can be directed to the corresponding author.

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
