# Peer review of "Innovative Antifungal and Food Preservation Potential of Eucalyptus citriodora Essential Oil in Combination with Modified Potato Peel Starch"

_foods, 2025, doi:10.3390/foods14040602_

Round 1
Reviewer 1 Report
Comments and Suggestions for Authors
The author can find the comments below.
1. The essential oil composition, as evaluated by GC-MS, explicitly shows that some essential oils are present in high amounts within the mixture, such as Citronellal (58%), Eucalyptol (4.6%), Citronellol (~14%), and Caryophyllene (8%). Is it worth experimenting to compare these findings with previous literature where these essential oils were used as controls on the same microbes or similar ones? This could help determine whether the synergistic effect is significant for achieving antimicrobial activity. Have the authors considered using such experiments, given that these essential oils are commercially available and can also be purified from natural sources?
2. The section, "results" must be integrated with the section, "discussion", as it is currently hard to follow the discussion in relation to the results. Please merge them to improve the readability of the paper.
3. The authors studied the color of starch, but its significance in the context of the current study is underestimated. Please provide more context.
4. The term "modification of starch" seems to imply chemical modification, but it appears to be just starch preparation. Please consider simplifying this to avoid confusing the readers.
5. FTIR studies of native and autoclaved potato-peeled starch show almost no difference in chemical functionalization between the two. In my opinion, the overall starch molecular structure remains the same, resulting in similar FTIR chemical information (stretching and bending). As a suggestion, please consider using XPS, Raman, or other molecular spectroscopy techniques to better understand the molecular arrangement rather than just chemical functional groups.
6. Please explain the CH bending of starch in relation to its crystalline and amorphous forms. What form of glucose CH bending are the authors referring here? Explain how one form of glucose within the starch can be differentiated from the other in terms of crystallinity or amorphousness.
Author Response
Author’s response to reviewer’s comments
Title: Innovative antifungal and food preservation potential of Eucalyptus citriodora essential in combination with modified potato peel starch
Dated 07-02-2025
Dear Editor,
The authors would like to thank the Editor and the reviewers for valuable suggestions and encouraging remarks. We have revised the manuscript as suggested by the reviewers and the Editor. Kindly find responses to reviewer’s comments in detail below. In the revised manuscript, corrections are made in red colour text.
Reviewer’s 1 comments:
- The essential oil composition, as evaluated by GC-MS, explicitly shows that some essential oils are present in high amounts within the mixture, such as Citronellal (58%), Eucalyptol (4.6%), Citronellol (~14%), and Caryophyllene (8%). Is it worth experimenting to compare these findings with previous literature where these essential oils were used as controls on the same microbes or similar ones? This could help determine whether the synergistic effect is significant for achieving antimicrobial activity. Have the authors considered using such experiments, given that these essential oils are commercially available and can also be purified from natural sources?
Response: Thank you for the valuable suggestion, the mentioned components were all present within Eucalyptus citriodora EO and the corresponding %age indicates the relative peak area of each component present within the EO. The discussion section is now enriched with the information as advised by the reviewer by comparing with the previously published literature indicating the effect of EOs against similar microbes (Lines 372-378). However, the authors did not attempt to isolate each component of EO and test separately. Due to lack of in-house facility authors didn’t incorporate the isolation of each component of EO, however this was reflected in the limitation of the study at the end of discussion section.
- The section, "results" must be integrated with the section, "discussion", as it is currently hard to follow the discussion in relation to the results. Please merge them to improve the readability of the paper.
Response: Thank you for your feedback. However as per journal policy, discussion was segregated from the results. Yet the discussion is in the same sequence as that of result without headlines. The journal format required to separate the discussion section as stand alone.
- The authors studied the color of starch, but its significance in the context of the current study is underestimated. Please provide more context.
Response: As advised by the reviewer 1, the significance of starch color in the context of the current study is broaden under the discussion section of the revised manuscript (Lines 342-348).
- The term "modification of starch" seems to imply chemical modification, but it appears to be just starch preparation. Please consider simplifying this to avoid confusing the readers.
Response: We are thankful for the reviewer’s observation regarding the term “modification of starch”. The process of autoclaving induced significant structural and physicochemical alterations which justifies the term “modification”. The color alteration reflects the modifications indicating internal rearrangement such as degradation of polysaccharide chains, formation of reducing sugars and their associated by-products or caramelization due to high temperature treatments. Furthermore, morphological changes such as granule deformation, fragmentation, or surface erosion also indicate the structural alterations (which was supported by microscopic observation of starch granules). In FTIR spectra, although there was no major shift observed, yet changes in peaks intensity suggested physical modifications.
- FTIR studies of native and autoclaved potato-peeled starch show almost no difference in chemical functionalization between the two. In my opinion, the overall starch molecular structure remains the same, resulting in similar FTIR chemical information (stretching and bending). As a suggestion, please consider using XPS, Raman, or other molecular spectroscopy techniques to better understand the molecular arrangement rather than just chemical functional groups.
Response: We agree with the reviewer that there was no significant difference in the chemical functionalization observed between native and modified starch. Autoclaving is a physical modification method in which starch is treated at high temperature and pressure. Therefore, this process particularly induces physical changes such as granular deformation, color changes or gelatinization (as supported by the microscopic evaluation). In the current investigation, formation of gelling texture in the modified starch-based coatings was observed which was consistent with the physical changes induced by autoclaving (data not shown in this study).
Difference in chemical functionalization could be observed when the starch is primarily subjected to chemical modifications (acetylation, crosslinking, oxidation or etherification) which results in the loss of previous bands or addition of new bands. As there was no new functional group introduced and the consistent peaks suggested that the fundamental chemical structures of native and modified starch majorly remained unchanged during autoclaving but primarily induced physical modifications. This remained evident particularly through changes in peak intensity only. The results of this study are supported by recently published literature which can be accessed in following studies:
- 1016/j.heliyon.2023.e19581
- https://doi.org/10.1016/j.ijbiomac.2018.08.066
With reference to the proposed analytical techniques, we apologize as the recommended analytical facilities are not available in house, therefore we have highlighted this in future perspective at the end of discussion section.
- Please explain the CH bending of starch in relation to its crystalline and amorphous forms. What form of glucose CH bending are the authors referring here? Explain how one form of glucose within the starch can be differentiated from the other in terms of crystallinity or amorphousness.
Response: Thank you for valuable comments, it was actually C-OH vibration instead of C-H vibration, the figure is revised and typographical mistake is corrected in the revised manuscript. Moreover, following discussion is added in the revised manuscript (L260-265) to support the concerns raised by the reviewer. The concept of modification is now justified by support from previously published literature.
(L261-266) Peaks between 980-1040 cm-1 correspond to anhydrous glucose ring and major peak shifts were observed within this region after the modification of starch [18]. There was a sharp decrease in the intensity of peak at 998 cm-1 after the modification which was attributed to change in structural configuration of starch. It was previously reported that changes in the configuration of glycosidic linkage was responsible for variations in IR region in the range of 920-1000 cm-1 [29].
Reviewer 2 Report
Comments and Suggestions for Authors
The paper presents a valuable contribution to the field of food preservation and sustainability by exploring an innovative combination of Eucalyptus citriodora essential oil (EO) and modified potato peel starch (M-PPS) for antifungal activity.
Could the title be made more concise and focused on the novelty of the study? For example, would a title like "Innovative Antifungal Coating from Eucalyptus Essential Oil and Modified Potato Peel Starch" better reflect the unique contribution of this research?
Can the abstract more clearly highlight the novelty of the study and include specific numerical findings, such as the percentage of fungal inhibition or MIC values, to provide a clearer summary for readers?
Could you elaborate further on the gap in knowledge this study addresses, particularly in the use of natural, eco-friendly coatings for food preservation?
Would adding references to recent studies on biodegradable coatings and essential oils strengthen the background section?
Have all relevant recent studies on antifungal properties of Eucalyptus essential oil and modified starch-based coatings been included to support your objectives?
Could you provide additional details about the statistical replicates used in experiments, particularly for the preservation studies with peanuts and rice?
In the section "Preservation of peanuts by modified starch and EO-based coatings," would it be helpful to clarify the specific concentrations used for the control groups?
Could the figures be improved with clearer labels, legends, and units (e.g., in Figures 6 and 7) to enhance readability?
In results tables like the GC-MS analysis, would highlighting the significance of citronellal’s predominance help readers focus on the key findings?
Have you compared your findings, such as the MIC values for Eucalyptus EO, to similar studies in the discussion? Would this provide more context for your results?
Can you elaborate further on the mechanisms of antifungal action observed, with references to how essential oils disrupt fungal membranes or interact with cell walls?
Would discussing the broader industrial applications of this preservation system provide more relevance to the study?
Also, I noticed several gaps in methodology for this paper:
Lines 211–213 Specify whether the replicates were biological or technical. Add a power analysis to validate the sample size and ensure the results are statistically robust.
Lines 197–198 Include controls with synthetic antifungal agents or unmodified starch coatings to better contextualize the performance of the proposed coating system.
Lines 192–194 Provide information on how coating thickness, homogeneity, and stability were optimized.
Lines 206–208 Include tests under real-world storage conditions such as varying temperature, humidity, and time periods.
Lines 366–367 Quantify the release kinetics of EO over time to validate the sustained release claim.
Author Response
Author’s response to reviewer’s comments
Title: Innovative antifungal and food preservation potential of Eucalyptus citriodora essential in combination with modified potato peel starch
Dated 07-02-2025
Dear Editor,
The authors would like to thank the Editor and the reviewers for valuable suggestions and encouraging remarks. We have revised the manuscript as suggested by the reviewers and the Editor. Kindly find responses to reviewer’s comments in detail below. In the revised manuscript, corrections are made in red colour text.
Reviewer’s 2 comments:
- Could the title be made more concise and focused on the novelty of the study? For example, would a title like "Innovative Antifungal Coating from Eucalyptus Essential Oil and Modified Potato Peel Starch" better reflect the unique contribution of this research?
Response: Thank you for the valuable suggestion. The title has been modified as suggested and the novelty of the study is discussed under the section “Introduction” (Lines 85-89)
- Can the abstract more clearly highlight the novelty of the study and include specific numerical findings, such as the percentage of fungal inhibition or MIC values, to provide a clearer summary for readers?
Response: The novelty of the study as well as numerical findings such as the percentage of fungal inhibition or MIC values are incorporated in the revised abstract.
- Could you elaborate further on the gap in knowledge this study addresses, particularly in the use of natural, eco-friendly coatings for food preservation?
Response: The gap in knowledge addressed in this study is highlighted in the revised manuscript under the section “introduction”
(L55-58) For the sustainable and economical production of starch, beyond the conventional sources (corn, potatoes, rice, wheat and pulses), there is a gap in understanding the potential of non-conventional sources, particularly agrifood waste.
(L85-89) Despite the growing interest in the natural preservation systems, to the best of our knowledge, there is very limited data available on the development of modified potato starch and E. citriodora EO based preservation systems. Our study addresses a significant gap in the literature and aims to investigate the potential of combining E. citriodora EO and modified potato peel starch to create an innovative preservation system.
- Would adding references to recent studies on biodegradable coatings and essential oils strengthen the background section?
Response: As advised, recent studies on biodegradable coatings and essential oils are added in the background information to keep the readers up to date and reinforce the claim made by the authors.
https://doi.org/10.1002/pts.2787
https://doi.org/10.1016/j.foodchem.2021.130671
- Have all relevant recent studies on antifungal properties of Eucalyptus essential oil and modified starch-based coatings been included to support your objectives?
Response: To the best of our knowledge, we have consulted all the relevant recent studies. However, there is very limited data available on the development of modified potato peel starch and E. citriodora EO based preservation systems. The discussion section includes the comparison with similar preservation system based on EOs to support the findings of the current study.
- Could you provide additional details about the statistical replicates used in experiments, particularly for the preservation studies with peanuts and rice?
Response: Thank you for the suggestion, All the experiments were performed in triplicates, the suggested information is incorporated into the revised manuscript in the respective sections as advised by the reviewer. Moreover, the statistical analysis is revised to provide more insight about the number of replicates in each experiment.
“Statistical analysis
All experiments were performed in triplicates (except starch color analysis, per-formed in 5 replicates) and significant differences (p < 0.05) among mean observations were highlighted by using one way analysis of variance and Tukey’s HSD tests. The instrumental analysis such as FTIR, TGA, and GCMS were performed once.”
In the section "Preservation of peanuts by modified starch and EO-based coatings," would it be helpful to clarify the specific concentrations used for the control groups?
Response: As advised, the specific concentrations used for the control groups are further clarified in the revised manuscript (L220-223).
Could the figures be improved with clearer labels, legends, and units (e.g., in Figures 6 and 7) to enhance readability?
Response: Thank you very much for the valuable suggestions, the figure legends are revised and information is clarified for readers.
In results tables like the GC-MS analysis, would highlighting the significance of citronellal’s predominance help readers focus on the key findings?
Response: As per the suggestion, the predominance of citronellal’s is highlighted by adding asterisk symbol (*) in the table along with table footer “*The component present in highest concentration (in terms of peak area)”.
Have you compared your findings, such as the MIC values for Eucalyptus EO, to similar studies in the discussion? Would this provide more context for your results?
Response: Thank you for the valuable insight, the findings of this study are compared with previously published literature in the discussion section of the manuscript. The MIC values and EO are compared to similar studies within the revised manuscript, Lines 385-390.
Can you elaborate further on the mechanisms of antifungal action observed, with references to how essential oils disrupt fungal membranes or interact with cell walls?
Response: The antifungal mechanism of action is explained in discussion section of manuscript, lines 390-390.
Would discussing the broader industrial applications of this preservation system provide more relevance to the study?
Response: The suggested information as advised by the reviewer is reflected in the discussion section of the manuscript, lines 425-430.
Detailed Reviewers Comments:
Lines 211–213 Specify whether the replicates were biological or technical. Add a power analysis to validate the sample size and ensure the results are statistically robust.
Response: The statistical analysis section is now revised and information is clarified for the readers,
“All experiments were performed in triplicates of independent experiments (except starch color analysis, per-formed in 5 replicates) and significant differences (p < 0.05) among mean observations were highlighted by using one way analysis of variance and Tukey’s HSD tests. The instrumental analysis such as FTIR, TGA, and GCMS were performed once.”
Lines 197–198 Include controls with synthetic antifungal agents or unmodified starch coatings to better contextualize the performance of the proposed coating system.
Response: We are obliged for the valuable suggestion. However, starch in its native form possesses restricted functionalities such as poor gelling properties (data not shown), therefore coating system was developed by using modified starch. That’s why we did not include native starch for the formulation of coating system.
Lines 192–194 Provide information on how coating thickness, homogeneity, and stability were optimized. Lines 206–208 Include tests under real-world storage conditions such as varying temperature, humidity, and time periods. Lines 366–367 Quantify the release kinetics of EO over time to validate the sustained release claim.
Response: Thank you for your valuable insight. Unfortunately, authors did not incorporate the these experimental parameters suggested by the reviewer, however keeping in view the valuable suggestion of the reviewer, the suggested parameters are presented as the limitation of the current investigation under the “Discussion” section.
Reviewer 3 Report
Comments and Suggestions for Authors
The article "Antifungal and food preservation potential of Eucalyptus citriodora essential in combination with modified potato peel starch" describes the antimicrobial activity of an essential oil that can be used for food packaging in combination with starch extracted from agri-wastes. While the article shows promise, it requires major revisions before it can be considered for publication in this journal.
The English language needs some polishing for style and typos.
Eucalyptus citriodora essential oil has a similar composition with citronella essential oil (lemon grass); thus comparisons with previous literature where citronellal, citronelol etc were encapsulated in packaging films like previous work of Motelica et al on essential oils encapsulated in polysaccharides films can support the idea of this manuscript.
Thermogravimetric analysis was done in air or inert atmosphere? If it is done in inert atmosphere the residual mass can be explained as a carbonaceous amorphous residue.
Which are the strengths and the limitations of this study? Changing the starch source like, from banana peel will significantly alter the results?
Mechanical properties of packaging films are important and should be determined, unless authors intend to produce this as a thin film, as a coating. Also the barrier properties are important in all cases (permeability to water vapour and gases like oxygen that can promote the growth of various microbes and lead to alteration of nutrients and vitamins).
Encapsulation efficiency, loading capacity and release profile should be determined for the obtained films.
Please add some more studies about the antimicrobian effects and biocompatibility of essential oils. You can see for example doi: 10.3390/pharmaceutics16091225.
The conclusion should reflect the heuristic of the study. How is this system a better one? Conclusion section must be reworked to underline the novelty and advantages of this research, with actual numbers. The MIC 3.125 from Conclusion section is missing measurement units. The conclusion part does not highlight the salient findings and future perspective.
Author Response
Author’s response to reviewer’s comments
Title: Innovative antifungal and food preservation potential of Eucalyptus citriodora essential in combination with modified potato peel starch
Dated 07-02-2025
Dear Editor,
The authors would like to thank the Editor and the reviewers for valuable suggestions and encouraging remarks. We have revised the manuscript as suggested by the reviewers and the Editor. Kindly find responses to reviewer’s comments in detail below. In the revised manuscript, corrections are made in red colour text.
Reviewer’s 3 comments:
The English language needs some polishing for style and typos.
Response: The manuscript is now thoroughly checked and language errors/typographical mistakes are rectified.
Eucalyptus citriodora essential oil has a similar composition with citronella essential oil (lemon grass); thus comparisons with previous literature where citronellal, citronelol etc were encapsulated in packaging films like previous work of Motelica et al on essential oils encapsulated in polysaccharides films can support the idea of this manuscript.
Response: We are obliged for the helpful recommendation. The following studies are consulted to strengthen the concept (introduction section)
https://doi.org/10.3390/nano11092377
https://doi.org/10.3390/pharmaceutics16091225
Thermogravimetric analysis was done in air or inert atmosphere? If it is done in inert atmosphere the residual mass can be explained as a carbonaceous amorphous residue.
Response: We apologize for the oversight. Thermogravimetric analysis was done in inert atmosphere. The manuscript is revised and corrected.
Which are the strengths and the limitations of this study? Changing the starch source like, from banana peel will significantly alter the results?
Response: The strengths of the study are reported at the end of “introduction” section
For the sustainable and economical production of starch, beyond the conventional sources (corn, potatoes, rice, wheat and pulses), there is a gap in understanding the potential non-conventional sources, particularly agrifood waste.
Despite the growing interest in the natural preservation systems, to the best of our knowledge, there is very limited data available on the development of modified potato starch and E. citriodora EO based preservation systems. Our study addresses a significant gap in the literature and aims to investigate the potential of combining E. citriodora EO and modified potato peel starch to create an innovative preservation system.
The limitations of the study are highlighted at the end of “Discussion” section
Despite the current investigation presented a promising food preservation system, certain limitations need to be addressed to improve the industrial applicability of the developed coatings. Future studies should explore coating parameters such as thickness, homogeneity and stability to obtain consistent coating quality. In this study, the coatings were not evaluated in real world storage conditions such as fluctuation in temperature, humidity levels and storage time period, which might act as critical parameters in the efficacy of food preservation system. Moreover, the coatings were aimed for direct application on food surfaces. However, future studies could explore the application of M-PPS and E. citriodora EO based thin films and assess mechanical properties (tensile strength) and barrier properties (oxygen permeability, water permeability). Another key limitation is the evaluating the encapsulation efficiency, loading capacity and release kinetics of EO, confining understanding of sustained release characteristics. Addressing these gaps in future studies will pave ways to enhance the practicality and applicability of eco-friendly coatings for broader commercial use.
Yes, the results will be significantly altered by changing the source of the starch.
Mechanical properties of packaging films are important and should be determined, unless authors intend to produce this as a thin film, as a coating. Also the barrier properties are important in all cases (permeability to water vapour and gases like oxygen that can promote the growth of various microbes and lead to alteration of nutrients and vitamins).
Response: In the current investigation, the coatings were aimed for direct application on food surfaces. However, future studies could explore the application of M-PPS and E. citriodora EO based thin films and assess mechanical properties (tensile strength) and barrier properties (oxygen permeability, water permeability) which are critical parameters for microbial growth and nutrient preservation. This is represented as the limitation of the study and mentioned at the end of “Discussion” section.
Encapsulation efficiency, loading capacity and release profile should be determined for the obtained films.
Response: Thank you for your valuable insight, however we did not include the mentioned parameters in the scope of current study. However, based on valuable suggestion of the reviewer, the suggested parameters are presented as the limitation of the current investigation under the “Discussion” section
Please add some more studies about the antimicrobian effects and biocompatibility of essential oils. You can see for example doi: 10.3390/pharmaceutics16091225.
Response: Thank you for your valuable suggestion. The suggested study was consulted and cited in the revised manuscript.
The conclusion should reflect the heuristic of the study. How is this system a better one? Conclusion section must be reworked to underline the novelty and advantages of this research, with actual numbers. The MIC 3.125 from Conclusion section is missing measurement units. The conclusion part does not highlight the salient findings and future perspective.
Response: The conclusion section is thoroughly revised as advised by the reviewer and strengthened by including the main points advised.
“EO of E. citriodora has strong antifungal effect at very low concentration with MIC 3.125 µl/ml against two major foodborne fungi (P. griseofulvum and A. niger). EO compromised the cellular integrity of fungi and released the cellular constituents. E. citriodora EO along with modified starch-based coating inhibited the growth of fungi in peanuts artificially inoculated with fungi. The high cost of natural preservative systems is always an obstacle in sustainable application of food preservation systems. Potato peel starch due to its low cost and E. citriodora EO due to its abundance can be used to formulate a low cost and sustainable food preservation system for food and feed industry.”
Round 2
Reviewer 2 Report
Comments and Suggestions for Authors/
Reviewer 3 Report
Comments and Suggestions for Authors
The article seems well-revised and now can be recommended for publication.